# The Parole Officials' Views Concerning the Parole System in Rehabilitating Offenders: Experiences from Brits Community Correction Centre in South Africa

**Olebogeng Wendy Letlape** [1] **and Misheck Dube** [2,*]

1    Department of Social Work, Faculty of Health Sciences, North-West University, Mafikeng 2745, South Africa; olebogengletlape@gmail.com
2    Department of Social Work, Lifestyle Diseases Entity, Faculty of Health Sciences, North-West University, Mafikeng 2745, South Africa
*    Correspondence: misheck.dube@nwu.ac.za; Tel.: +27-183-892893

**Abstract:** Unless the views of parole officials are unpacked and understood with regard to rehabilitating offenders in correctional centres with limited resources in South Africa, there will be less effort devoted towards ensuring the effectiveness of the parole system. This paper captures the working experiences of the parole officials at Brits Community Corrections in South Africa with regard to the use of parole in the rehabilitation of offenders. Using a qualitative approach and an exploratory descriptive design, the study employed total population purposive sampling to ensure the inclusion of all parole officials in the study to provide their working experience. In-depth interviews that provided data saturation with four parole officials were analysed thematically, critically discussed and compared to existing literature. The major findings were that the challenges faced by parolees in the communities were the main problems preventing the successful implementation of parole services. This led to endemic frustrations among the parole officials in the execution of parole services in the Brits area. This paper recommends relevant holistic approaches as interventions to improve the parole systems in the area and improve the working experiences of parole officials.

**Keywords:** parole; parole officials; parole system; rehabilitating offenders; Brits

## 1. Introduction

The parole system varies across nations, although there are many similarities in the fact that parole brings about the social reintegration of offenders to their societies with the assumption that they have become responsible citizens. This study explores the views of professionals who are directly involved in the processes of parole at Brits Community Correctional Centre in South Africa.

Parole as a means of rehabilitating offenders is not only a South African concept but is also practised globally. One of the countries in Europe with outstanding parole practices is Sweden. In Sweden, parole refers to the required release and community supervision of offenders, once two-thirds of their prison sentence is served (Basset 2016). The Swedish parole system adopts a strategy of having layman supervisors who volunteer to help monitor and give social support to the offenders on a weekly basis. This helps to reduce the workload and ensures that parolees and probationers receive support (Basset 2016). Research has shown that the Swedish correctional values aim to be resistant to future social and political changes by maintaining long-term approaches to rehabilitation and detention (Basset 2016). It is important to note that parole release does not mean one is completely free; it is still part of the crime sentence to ensure that one is integrated back into the community as a better citizen. Therefore, there is supervision and monitoring to ensure that the offender abides by the law and does not reoffend. Sweden, like other countries,

makes use of electronic monitoring to track offenders and to ensure that they comply with their parole conditions.

In terms of the use of technology, Sweden as a country has shown evidence that the electronic monitoring of offenders serves the ultimate purpose of parole, which is to ensure that the offender is reintegrated into society because it is believed that they are rehabilitated (Basset 2016). All efforts made to ensure that a parolee is well monitored in the community are there to reduce the rate of recidivism and to ensure that society is safe. Basset (2016) furthermore shares that Sweden is far ahead of many countries in the electronic monitoring of offenders and it has yielded the best results in minimising the issue of recidivism. The impact of the adopted approach in Sweden has been resounding. Due to the success of reducing criminality since 2004, Sweden closed 56 prisons and reduced the number of prisoners from 5722 to 4500 out of a population of 9.5 million people (James 2014). Whilst there is lack of recent data on the phenomenon, analysts have predicted that the new regulations against terrorism and crime enacted in Sweden may result in the need for new prison spaces in the future (Saleem 2020).

Drifting away from Europe to Asia, Malaysia has demonstrated that parole can work to reduce reoffending among parolees. In the year 2020, the Prisons Department director-general, Datuk Sri Zulkifli Omar, reiterated that the Malaysian parole system has proven to be successful in reducing the rate of recidivism in Malaysia (Malay Mail 2020). In the year 2008, Hassan et al. (2017) shared that the government in Malaysia enforced the Prison Act of 1995, which clearly stipulated the eligibility criteria for parole depending on the severity of the crime committed by the offender. The Act excludes those who committed crimes such as murder, rape, incest and offences related to kidnapping and possession of firearms for eligibility to ensure that society is safe from dangerous criminals (Hassan et al. 2017).

In South Africa, parole is pinned on a needs-based approach, similar to other countries like Sweden. Through the needs-based approach adopted in South Africa, people who break the law are not viewed as criminals; rather, they are assumed and viewed as primarily people with needs who require help for their behaviour to be corrected (James 2014). This approach demonstrates that the underlying problems or factors that lead to criminal behaviour are understood and taken into consideration to try to eliminate those problems. According to the South African correctional system, which is guided by the White Paper on Corrections of 2005, Rule 66(1) of the Standard Minimum Rules:

> "To these ends, all appropriate means shall be used, including religious care in the countries where this is possible, education, vocational guidance and training, social casework, employment counselling, physical development and strengthening of moral character, in accordance with the individual needs of each prisoner, taking account of his (or her) social and criminal history, his (or her) physical and mental capacities and aptitudes, his personal temperament, the length of his (or her) sentence and his (or her) prospects after release."

The South African approach to parole has been harnessed by the writings of Mujuzi (2011, p. 206), who highlights that parole is about "rewarding offenders for complying with their sentence plan and participating in rehabilitation programmes and combating recidivism by ensuring the gradual re-integration of offenders" into communities. Offenders who served their sentence are not expected to reoffend and be problematic in their communities, as the process is intended to rehabilitate them to leave a life of crime and become law-abiding citizens. While that philosophy to parole is ideal, correctional centres have been criticised for struggling with managing the numbers of new cases, and with offenders who have been under correctional services for offending more than once, a term called recidivism (Zara and Farrington 2015). According to Zara and Farrington (2015), recidivism is the official criminal involvement of a person who, after being convicted for an offence, commits another crime. This consequently affects communities as they continually suffer from high rates of crime, exerting pressure on the government to devise ways to reduce crime and the rate of recidivism with strained resources.

Research has shown that recidivism rates vary worldwide. It was reported that Sweden had a 43% reconviction rate between 2005 and 2015, while in Norway, two-year recidivism rates increased from 14% to 42% from 2010 to 2011 (Fazel and Wolf 2015, p. 6). According to Khwela (2015, p. 407), general figures for recidivism indicate that 50% to 70% of inmates reoffend within a period of three years. Sadly, there is lack of specific statistics on the rate of recidivism in South Africa, despite the rate of crime that continues to rise. For example, the total number of crimes committed in Brits, in the North West Province, increased from 4856 reported crimes in 2016 to 9466 in 2017 (Crime Statistics South Africa 2017). This reported crime spike in one year in the Brits area of South Africa, despite the existence of a Community Correctional Centre, has drawn the attention of the researchers; hence this paper.

To clarify how parole works in South Africa, offenders serve a portion of their sentence in custody, and another portion outside custody as parolees in their communities, and are not expected to reoffend, especially when they are under parole, which falls under community correctional supervision (Chikadzi 2017). This means that the parole board needs to assess the suitability of an offender to be monitored for part of their sentence outside of custody (Department of Correctional Services 2018). The offender, who has now become a parolee, is sent into the community to finish off a portion of the sentence within the community. This is described as community corrections.

The purpose of community correctional supervision is to protect the community and to prevent recidivism by ensuring control and effective supervision of offenders (Department of Correctional Services 2018). There has been growing concern about the lack of readiness of communities in South Africa in accepting ex-offenders back into the communities, as they normally stigmatise them, thereby jeopardising the rehabilitation process and often resulting in reoffending (Chikadzi 2017).

An important research question was propelled this study. The question that prompted the study was what are the views and experiences of the parole officials regarding parole in rehabilitating and reintegrating parolees into the community in the Brits area? This question emanated from the fact that in the Brits area, there is a Community Correctional Centre that has a vibrant parole system, yet with high rates of reoffending among parolees (Crime Statistics South Africa 2017). With such a critical research question, this paper aims to analyse how the parole system in Brits assists in rehabilitating offenders and reintegrating them in the community. This analysis was carried out using the expert views of parole officials who were key informants, as they possess the expertise, experience, and knowledge needed to understand the phenomenon in question.

The findings of the study have highlighted the fact that parole officers have distinct roles that functionally complement each other in their execution of duties. This was lauded as a potential strength in the correctional centre's activities. On the contrary, the parole activities of the centre have been found to be marred by setbacks such as the stigmatisation of ex-offenders in the community, a lack of resources for parole officers, and drug abuse among parolees, thereby increasing the chances of negatively affecting the activities of the rehabilitation process (Huebner et al. 2019; IvyPanda 2020)

## 2. Materials and Methods

This study employed a qualitative research approach. Teherani et al. (2015) describe qualitative research as a systematic investigation of a social phenomenon in its usual setting and the researcher being in the centre of data collection by examining why events occur, and what those events mean to the participants.

The study employed a qualitative research design to seek an understanding of the experiences of the participants to explore and explain how the participants understood parole activities in the process of rehabilitating offenders (Haradhan 2018).

An exploratory descriptive design was adopted in the study with the aim of describing the phenomenon by taking the participants' perspective (Hunter et al. 2018). This was also consistent with the social constructivist paradigm to which this study is committed. Social

constructivism holds that people give and attach different interpretations to a phenomenon due to the different lenses they use to understand the phenomenon (Teater 2014). This is essential for the generation of substantial and insightful information needed in the correctional environments and is directly consistent with the exploratory descriptive design this study used.

### 2.1. Sampling Method Applied

The sampling method that was used in this study was total population purposive sampling to identify primary participants who had personal experiences of parole as parole officials (Canonizado 2021). Other sampling techniques could not ensure this. Total population purposive sampling implies choosing all available participants according to pre-selected criteria relevant to a particular research question as the population is too small to create a sampling frame (Etikan et al. 2016; Canonizado 2021). Due to the specialised nature of parole services and the limited resources in the Brits Correctional Centre, all four correctional officials who were responsible for parole services participated in this study. These included three parole officers and one social worker. This was also consistent with the inclusion criteria for participating in this study. However, this affected the generalizability of the research findings as the findings may not be generalized to other settings (Padgett 2017). Anney (2014) argues that purposive sampling and thick descriptions are sufficient to facilitate transferability. Other researchers posit that if the context is fully described, the person who wants to transfer the findings needs to be responsible and be able to determine how sensible it is to transfer the findings (My Dissertation Coach 2019; Analytics Simplified 2023).

### 2.2. Data Collection Technique

The researchers conducted in-depth interviews as a technique of collecting data from the participants, utilising an interview schedule as a data collection instrument (Kothari 2014; Stinger 2014). The interview questions were semi-structured to allow the interviewers to gather information that a strictly structured interview guide may not solicit from the participants. All four correctional officials participated in in-depth interviews to allow the collection of rich information that enabled a data saturation point.

As an ethical requirement, the researchers sought ethical approval with the ethics number NWU-00479-19-S1 from the North-West University Faculty of Health Sciences Research Ethics Committee to proceed with the study. The Department of Correctional Services of the North West Province in South Africa required the ethical clearance letter to grand permission to conduct the study in the correctional services department (Department of Correctional Services 2018).

The participants were identified and recruited into the study through holding information-sharing meetings whilst observing the required health protocols to prevent the spread of COVID-19, as the data collection process happened amidst the COVID-19 pandemic. All four identified participants signed consent forms voluntarily agreeing to participate in the study before the interviews were conducted.

The data were collected from the participants after five days on agreed times and dates for each of the participants, to avoid congesting the interview venue and prevent the spread of COVID-19. Each interviewee participated in the interviews at a specified time slot during which no other participants were available to maintain privacy. Consistent with ethics, all interviews in this study were recorded with consent from the participants. To allow the presentation of the data, the participants were named with alphabetical letters so that information may not be linked to a particular participant, ensuring anonymity (Kothari 2014), except for the social worker.

### 2.3. Data Analysis

According to Kothari (2014), data analysis occurs after the data have been collected and requires several closely related operations such as the establishment of categories, the application of these categories to raw data through coding and tabulation, and then

drawing textual inferences. Thematic data analysis was used in this study. Maguire and Delahunt (2017) share that thematic data analysis involves the process of identifying themes within qualitative data. The researchers transcribed the collected data by listening to the audio-recorded interviews to produce meaningful information; therefore, the raw data were classified into purposeful thematic categories, which guided the data presentation and analysis. Thereafter, the information was backed and compared with existing literature to create discussion points. The data analysis process followed a six-stage process summarised by Maguire and Delahunt (2017), involving becoming familiar with the data through the reading of the transcripts, generating initial codes through compressing the data into small chunks of meaning, searching for themes, reviewing themes, defining themes, and, finally, writing up the research report through a closely monitored process to prevent deviance from ethics. One researcher validated the process of creating the categories, whilst the ethics committee of the university ensured that there was no deviance from the approved research processes and ethical conduct by the researchers.

## 3. Findings

### 3.1. Professional Backgrounds of the Parole Officials

The study sought to establish the professional backgrounds of the participants to understand their professional profiles within the parole system. This part of the interview centred on their position in the parole system, their qualifications, and the number of years of experience they had working with the parole system. Table 1 below depicts what the study found from the participants.

**Table 1.** Professional backgrounds of parole officials.

| Participant Name | Position | Highest Qualification | Working Experience |
|---|---|---|---|
| Participant A | Social Worker | Social Work degree | 16 years |
| Participant B | Monitoring Officer | Education degree | 22 years |
| Participant C | Correctional Officer | Grade 12 | 10 years |
| Participant D | Monitoring Officer | Correctional Service Administration Certificate | 13 years |

Source: Letlape and Dube 2022.

The parole officials were asked about their professional background to better understand how they fit within the parole system and their experience (see Table 1). This information about the parole officials adds value and is important to ensure that the Community Corrections functions according to its mandate.

It is noteworthy that the parole officials have a critical role in the correctional services in South Africa. They need proper training and experience to carry out their mandated functions. When the parole officials were interviewed, they narrated their professional backgrounds, and the succeeding paragraphs captured the narratives, as follows:

*"Ok, I have done my normal Social Work degree at Potchefstroom when it was still university of CHO, then I started working in 2000. I was few months a community service worker and child welfare for one year and 6 years in NGO Welfare. I started to work at the end of 2006 at the Department of Correctional Services. I worked at the maximum facility first before I started working here [Brits Community Corrections]"* —Social worker

*"I have Matric. After my Matric, I have a BA degree and a post-graduate diploma in education which I obtained prior to joining the Department. So, I joined the Department in 2000, I started working at the prison until 2005 and since then I have been working here at Community Corrections working as a monitoring officer. I am responsible for probationers and parolees."* —Participant A

*"I am a correctional officer under monitoring, I have grade 12. I did policing, however I did not finish my studies. However, when I joined the correctional services, I got training offered by the Department. So I have experience when it comes to monitoring offenders."* —Participant B

*"My basic training in correctional services I did it in 1990, then from there I was working in the centre where I was guarding the offenders and taking them to do their commitments, like going to the courts or hospital. I was also in charge of the farms for some time until 1994 when I came to Losperfontein. Then in 1997 I was appointed by the commissioner to be a monitoring official in the community corrections. In 1999 I started doing Reintegration Management Case Supervisor (RMCS) and worked until now in 2012. My qualifications are grade 12 and I have correctional service admin. There were two modules which is 'Prison Matters' and 'Statistics Personnel and Finance'."* —Participant C

From the narratives of the parole officials, all officials had formal educational training backgrounds with more than three years of work experience within the Department of Correctional Services. In lieu of the training backgrounds of the officials, some researchers recommend that such training is needed within any correctional to assist the Department of Correctional Services with the proper execution of parole services and the prevention of recidivism (Magadze 2016; Paulson 2013).

All of the correctional officials have extensive experience of working within the parole system. This may indicate that the Department of Correctional Services has invested in the officials to execute its mandate. Such vast experience may assist in ensuring stability within the department.

### 3.2. Roles and Responsibilities

In the parole system, parole officials have specific roles to fulfil in their daily activities. The roles and functions come with specific experience that this paper explored during the interviews. The roles and the responsibilities of the of the parole officials aided in gaining an understanding of the phenomenon of parole and how it serves to ensure that offenders are rehabilitated. Below is what the key informants had to say about their roles and responsibilities at Brits Community Corrections:

*" . . . I do 'Anger Management' programme and 'Life Skills', so what I do, I concentrate on relationships, self-image that's inside life skills. I even give them training on entrepreneurial skills trying to teach them how a budget works because others don't. I do 'Substance Abuse' programme, those are the type of things I concentrate on psychosocial support."* —Social worker

*"My daily responsibilities are ensuring that all parolees and probationers who are to be released on a particular day, are called in and have an exit interview with them and discharge them from the system. Any other parolees and probationers who came to the office, it is my primary job to admit them into the system and to explain the parole conditions, as well as referring to the RMCS and social worker for further management."* —Participant A

*"My duties are to report to my supervisor and that's where I will get directives as to where I will be conducting monitoring according to the schedule used in the office, so we go to different places to go monitor the offenders. We have different categories of offenders from low risk to high-risk offenders, so it depends on that day which category of offenders we are monitoring; sometimes we can even monitor 50 offenders in a day. When I do monitoring, I meet the offenders to make them sign and I meet the caregiver to get collateral information on the offender's behaviour at home. If we find that the parolee is not behaving or breaking his parole conditions then we warn that person, however if that parolee is a risk, then we revoke him."* —Participant B

*"As a parole officer my duties is to see to it that the parolees do their parole sentence as prescribed. As I am a parole officer, I meet them monthly in consultations to check if they are right with their caregivers at home and to check how we can adjust their house arrest. We give them house arrest; however we give them time to go seek for work, time to go to church, however when the parolee is employed we adjust their house arrest. The house arrest works like this on a Monday, Wednesday and Friday we give them 7- 3 to seek for work, on a Tuesday and Thursday is full day arrest, they don't go anywhere. On Saturday we give them only 4 hours for shopping and on a Sunday only if they affiliate to a church or sport activity, we give them time, but we need proof of affiliation. With the employed we change adjust their house arrest to fit their work schedule. We also look into corrective measures maybe when the parolee does not comply then we can change their house arrest conditions and if the parolee violated their parole conditions, then we do investigation and we see that person is not suitable to be on parole then we refer report to the Parole Board, and they decide to revoke that parolee"* —Participant C

The verbatim responses of the parole officials significantly highlighted the different roles that they play in the parole system. It is noteworthy that these roles expose the parole officials to different experiences. The social worker's primary role involves providing psychosocial support. Other participants, such as Participant A, deal with orientation of the offenders when offenders are admitted into the community corrections, while Participant B handled practical monitoring of the offenders. The role differentiation of the parole officer provides diversity in the rehabilitation of the offenders (Basset 2016). This paper asserts that the different roles the parole officers play in the parole system provide different perspectives and views about the parole system, and how it works to aid the rehabilitation of the offenders. It is expected that such diverse functional areas provide numerous work-related challenges for the parole process.

### 3.3. Challenges Faced by Parolees

The parole officials critically evaluated the parole system in the Brits area and described the challenges parolees faced in the rehabilitation process. Contrary to many expectations, it is not only the challenges faced by parole officers that introduce challenges into their work, but also the challenges the parolees faced, which impinge on the effectiveness of the rehabilitation processes. For that reason, the parole officials described challenges parolees faced that had negative influences on the execution of their duties and responsibilities. They described the challenges as la ack of facilities, substance abuse, and socioeconomic challenges. These are described in the succeeding sections.

### 3.3.1. Lack of Facilities

The parole system is majorly dependent on the facilities and resources available for use by parole officials. As such, a lack of facilities to aid the rehabilitation of parolees seemed to cause a significant challenge to the work of the parole officers. From the findings of the study, some parolees had addiction problems, and were recommended for rehabilitation. The issue of the facilities became a discussion point in the study. From the interviews conducted with the social worker, it emerged that parolees lacked facilities and resources to undergo drug addiction rehabilitation processes in the Brits area. Below are the verbatim responses of the social worker:

*"The challenge is that there are no resources . . . in the beginning when I started working here, I was able to send people to SANCA [South African National Council on Alcoholism and Drug Dependence] in Klerksdorp if they have a substance abuse problem. It doesn't work like that anymore, there is no facility in the whole of North West that is taking them in if they come through government because remember they do not have money to pay. So, there is one facility in the area where they need to pay R3000 a month but my clients still cannot afford.*

*I did have contact with NA [Narcotics Anonymous] and AA [Alcoholics Anonymous], there is one at Haarties and they used to help me but the problem with them it is rehabilitated people, all of them used to be drug addicts, there is no consistency because they backslide,"* —Social worker

The reality of the rehabilitation of parolees with substance abuse problems in the Brits area seems to be deeply problematic, as recounted by the social worker's experience. Such experience is understood from the psychosocial support needed by the parolees and provided by the social worker. This is explained by social construction theory, whereby reality is constructed and can only be construed from the participants' reality (Teater 2014). The NGO sector in many communities seems to be very helpful in stepping in to assist under-resourced communities such as the Brits area. Magadze (2016) agrees that that the NGOs are helpful in assisting and giving support to offenders to ensure that they are fully rehabilitated. However, that is not the case in Brits, where most NGOs do not have the capacity to assist with drug rehabilitation processes because their staff are not qualified professionals who can provide holistic treatment for addiction and give proper support. This was found to have impinged on the successful implementation of the parole system by the parole officers in the Brits area.

### 3.3.2. Substance Abuse

It is also ironic that the Brits area does not have facilities for rehabilitating parolees with substance addiction issues, and at the same time, many parolees who abuse substances require rehabilitation services in the area. Substance abuse among the parolees was found to be one of the problems inherent in the successful implementation of the rehabilitation of offenders on parole. This problem emerged during the interview with the social worker. In lieu of the substance abuse problems among the parolees, the social worker revealed:

*"The biggest issue I see, especially with substance abuse that happens with a lot of offenders, is that they never quit using dagga when in prison, so they are already busy with the entry drug when they are released so it is easy to go back to 'nyaope' [local slang for substance mixed with dagga, heroin, and other unknown substances]. Then the clinics try to help by giving methadone and the offenders become addicted to the methadone and there are pharmacies that make methadone easily available in small packages for something like R30 so if they are not on 'nyaope' then they are on that,"* —Social worker

The extent of substance abuse was found to be a lamentable phenomenon in Magadze's (2016) research findings. Magadze (2016, p. 34) found, for example, that about "two in five male ex-offenders and three in five female ex-offenders released from correctional facilities reported a combination of physical, mental, and substance abuse problems". The author reiterated that even among those who are in good health, the issue of substance abuse can be a stumbling block to successful transition from correctional facilities to the community. The American Addiction Centers (2022) share that the availability of drugs is a major factor in substance abuse in the United States of America. Methadone is said to be easily accessible due to its reasonable price, and many offenders can afford methadone and tend to become addicted to it easily.

### 3.3.3. Socioeconomic Challenges

The role played by the socioeconomic challenges experienced by the parolees emerged as one significant factor and drawback to the successful rehabilitation of parolees. Whilst many offenders would undergo rehabilitation, the socioeconomic challenges such as poverty and unemployment forced the parolees in the Brits area to reoffend and face reincarceration. From the findings of this study, the parolees faced myriad socioeconomic challenges, which resulted in reoffending. This general recidivism pattern among parolees was captured by the verbatim responses of the parole officers during the interviews:

*"With COVID, there are no jobs, there is no facilities in North West, there is nowhere we can help with skills development to work. Poverty is also a reality so they do not come*

*from families that can take care of their basic needs, it is one of the reasons they go back to prison.*" —Social worker

"*Most challenges faced by the offenders is being released to a family that is financially struggling and we know that those parolees won't find a job immediately and that can lead the parolees to reoffend.*" —Participant B

"*The common challenge is acceptance in the community, those are the challenges when we integrate the offenders in the community . . . we sometimes find that the community is prepared to integrate with offenders then you find other offenders are not ready to integrate in the community. But sometimes we win through making an awareness though community service being done and the community can see that that person is a parolee and their serving the community. The other challenge is reoffending, they reoffend with serious crimes, the modus operandi is the same. Those who have stolen they steal, those who killed, kill. They repeat their previous offenses.*" —Participant C

Facing a myriad of social challenges in the communities of domicile can surmount the gains of rehabilitation among the parolees. From the excerpts, it can be summarised that unemployment, poverty, having no identity documentation, social stigma from people in the community, and recidivism are common socioeconomic issues among parolees in the Brits area. Other researchers (Ndike 2014; Hegger 2015; Lekalakala 2016) and policy documents (Department of Correctional Services 2005) in South Africa found that such challenging social circumstances need to be properly harnessed because they pose a significant threat to the gains of parole services. In this paper, we argue that unless deliberate and comprehensive programmes to address such issues in the Brits area are implemented, the parole services in the area will remain challenging for parole officials and recidivism will remain endemically problematic.

## 4. Discussion of Findings

In lieu of the findings of the study, this paper drew significant discussion points regarding the parole system. One significant finding was that the parole officials had various training backgrounds, much of which had little focus on the psychosocial aspects of crime prevention, mitigation, and rehabilitation; yet, the parole officials' primary work and responsibility is to rehabilitate parolees. This paper argues that parole officials need to have qualifications and training that is related to their primary role to match the skills required and their key functions. This argument draws credence from the Department of Correctional Services White Paper on Corrections of 2005, which acknowledges the lack of necessary skills in the new rehabilitation-centred correctional system in South Africa and the department's inability to match the skill set required with much reliance on on-the-job training and human resource development (Department of Correctional Services 2005).

The major discussion point that this paper holds is that contrary to the view that the work of parole officials is challenging due to work-related environmental factors, this study found that the major problem results from the negative experiences of the parolees in the communities into which they should be integrated. The communities were found to not readily accept parolees as they expose them to stigma. Further, poverty and unemployment where significant factors in the lack of desired parole outcomes by the parole officers. These factors were found to propel recidivism and reverse the gains of the rehabilitation process. From a disorganisation theory perspective, such community elements are possible predictors of criminal behaviour (Bond 2015). Further, ecosystems theory considers the hostile environment around the parolee as being influential for recidivism; hence, stigma, as a hostile response from the community towards the parolee, can be linked to recidivism (Paulson 2013).

Further, drug addictions among parolees coupled with the lack of adequate rehabilitation centres crippled the parole officials' effort to properly rehabilitate offenders in the Brits area. The study found that many parolees had problems with drug addictions, yet there was a lack of proper rehabilitation centres to send them for rehabilitation. This was found

to be a significant drawback in the work experiences of the parole officials and a lack of proper standards for rehabilitation of offenders to prevent recidivism (Basset 2016).

## 5. Limitations of the Study

One limitation of this study was the low number of participants involved. This was the case even though the study included all of the parole officials. There were fewer parole officers at the Brits Community Correctional Centre. To avert this, the study conducted in-depth interviews with key informants to ensure that rich information was solicited from the participants and the data saturation point was reached. While the number of participants seems to have implications for the generalisability of the findings, the study findings are extremely important for poorly resourced community corrections and similar settings and should draw research interest.

Another limitation of the study was that the venue that the researchers used to interview the participants had a lot of restrictions due to the COVID-19 pandemic. However, proper arrangements and interview times were scheduled such that the spread of COVID-19 was minimised and the World Health Organisation Health protocols were observed.

## 6. Recommendations

In line with the findings of the study, the following recommendations have been made:

With the challenge of unemployment, the Department of Correctional Services should prioritise building a strong relationship with companies that can absorb qualified parolees and probationers without the criminal record being a hurdle.

To lessen the rate of crime, the Department of Correctional Services should adopt an element or division of punishment for more serious crimes to deter those who have not yet offended so as to lessen the rate of crime.

More non-governmental organisations (NGOs) should be encouraged to partner with community corrections and to adopt their objectives to establish a more holistic approach to societal responsibility for the rehabilitation of offenders.

Finally, more awareness programmes, such as: 'an offender is still my brother or sister', should be developed to curb the rate of stigma and labelling that offenders receive that has hampered the reintegration of parolees into the Brits community. There is need for families of offenders and the community at large to play an active role in assisting and supporting the reintegration of parolees.

**Author Contributions:** Conceptualization O.W.L. and M.D.; methodology, O.W.L. and M.D.; formal analysis, O.W.L. and M.D.; investigation, O.W.L.; resources, O.W.L. and M.D.; data curation, O.W.L. and M.D.; writing—original draft preparation, O.W.L.; writing—review and editing, M.D.; visualization, M.D.; supervision, M.D.; project administration, O.W.L. and M.D. All authors have read and agreed to the published version of the manuscript.

**Funding:** This research received no external funding.

**Institutional Review Board Statement:** The study was conducted in accordance with the Declaration of Helsinki, and approved by the Institutional Review Board (or Ethics Committee) of the Faculty of Health Sciences Research Committee (HREC) of the North-West University (NWU-00479-19-S1, date of approval-26 November 2020).

**Informed Consent Statement:** Informed consent was obtained from all participants involved in the study. Informed consent was signed for all the participants prior to conducting the study as required by the ethics committee of the university.

**Acknowledgments:** We acknowledge the support given by the participants in the study. Despite their busy work schedules, the participants availed themselves for the interviews with the researchers.

**Conflicts of Interest:** The authors declare no conflict of interest.

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
