# Peer review of "The Parole Officials’ Views Concerning the Parole System in Rehabilitating Offenders: Experiences from Brits Community Correction Centre in South Africa"

_socsci, doi:10.3390/socsci12070410_

Round 1
Reviewer 1 Report
The paper has many grammatical and punctuation errors, and much of it is confusing to read. And, I believe it is a very significant problem that there are only four participants in the sample. This calls into question the external validity of the findings.

The paper has many grammatical and punctuation errors, and much of it is confusing to read.
Author Response
Dear Reviewer
Thank you very much for the constructive feedback. We have rectified the manuscript issues according to your feedback. Refer to the attached correction sheet.
Kind regards;
Author

Reviewer 2 Report
In terms of the theoretical framework, it was important to update the state of the art, including recent studies and publications.
In methodological terms, it was important to attend to:
1. Interview is not a method bat a data collection technique (that fits into psychosocial survey)
2. The validation of the category system (and the way of doing that) is not clear. There was another coder experiencedin qualitative analysis techniques?
3. There was no schematic and descriptive-explanatory presentation of the category system (and eventually subcategories), which woul made the results presentation considerably clearer.

Author Response
Dear Reviewer
Thank you very much for the constructive feedback that improved pour manuscript. We have rectified the issues according to your feedback. Please refer to the attached correction sheet for detailed responses.

Round 2
Reviewer 1 Report
The revisions look good.
Reviewer 2 Report
I think there was a revision in order to improve the paper.